# HUMAN-LIKE COMMUNICATION STRATEGIES FOR IMPROVED MULTI-AGENT REINFORCEMENT LEARNING

## ABSTRACT

Multi-Agent Reinforcement Learning (MARL) has seen significant progress in recent years, enabling multiple agents to coordinate and optimize their actions in complex environments. However, integrating effective communication protocols into MARL frameworks remains a challenge, as it introduces issues such as increased state space dimensionality, lack of stationarity, and the need for interpretability. Inspired by human communication, which relies on prior knowledge, contextual awareness, and efficient information exchange, we propose a novel framework for incorporating human-like communication strategies to enhance the learning process. Motivated by recent advancements in natural language processing (NLP), multi-modal AI and object detection, we use text-to-mask models and human feedback to learn compact and informative communication strategies that facilitate coordination among agents to improve the overall performance. We demonstrate the efficiency of our approach on various multi-agent tasks and provide insights into emergent communication behaviors observed during training.

## 1 INTRODUCTION

Multi-Agent Reinforcement Learning (MARL) is a widely studied subject, where multiple agents interact with a shared environment, learning to optimize their return. While typically, each agent operates according to its own experience (i.e., decentralized joint-policy), many potential real-life applications allow the agents to communicate. By enabling agents to exchange critical information relevant to their shared task, communication can potentially enhance overall performance and coordination. However, integrating communication into multi-agent learning-frameworks introduces unique challenges; the communication increases the dimensionality of the state-space, and learning how to both control and communicate results in an increased non-stationarity. Previous research in the field can be divided to methods that utilize various aspects of centralized learning (Foerster et al., 2016; Sukhbaatar et al., 2016; Lowe et al., 2017; Jaques et al., 2019), use reward shaping to encourage communication (Jaques et al., 2019; Eccles et al., 2019), or harness special architectures to model the communication (Jiang & Lu, 2018; Lin et al., 2021; Lo et al., 2023). While these approaches benefit the learning process and show good empirical performance, the experiments are usually done in simple environments, in which either the optimal control or the optimal communication policies are relatively simple. In complex settings, current methods still perform suboptimally, or require an infeasible amount of interactions with the environment.

We aim to address complex problems. To better grasp the difference between a *complex* and a *simple* environment, consider the following examples. A navigation task in which one agent, the 'navigator', has to find a path to a control panel, which is always at the same location, from an arbitrary starting point, then press on either the blue button or the red button. Another agent, the 'dispatcher', knows on which button, red or blue, should the 'navigator' press. This is a *simple* task, for two reasons: (1) The information to communicate is stationary and (2) the communication affects only a single decision, when choosing on which button to press. With or without communication, the 'navigator' has to learn how to navigate to the control panel. If the location of the control panel is initiated randomly and the 'dispatcher' knows it, there is an added layer of complexity, since the entire navigation becomes communication-dependent. When dealing with a *complex* task, it is accepted to decouple the control-policy from the communication-policy; in the context of the example, the varying location of the control panel translates to a *complex* control task but a *simple* communication task, as the location of the control panel and the correct button stays fixed. If the

location of the control panel may change during an episode, the communication task would become even more challenging. We formulate a proper definition for such problems and provide further insights at Section 3.

When considering communication, it could be beneficial assimilating to human beings, as human communication is used daily to solve complex tasks. But while human-like communication protocols addressed for humans, it is not clear whether artificial agents that use RL would actually benefit from them. Nevertheless, the interpretability of such protocols may allow additional benefits, such as teaming up with humans or learning from them. This concept has been previously studied (Lazaridou et al., 2016; Havrylov & Titov, 2017; Karten et al., 2023), although in most cases, the main focus revolves around the emergence of language in rather simple control tasks, or involves a complex, task-specific learning schemes. Moreover, it has been shown that a true human-like communication is less likely to emerge naturally (Kottur et al., 2017). Another approach would be to learn from humans how to communicate, via behavioral cloning (BC) or RL from human feedback (RLHF), but in the face of a complex task it requires a vast amount of human-feedback which is hard to collect.

Drawing inspiration from human communication, which relies on prior knowledge, contextual understanding, and efficient information exchange, there exists an opportunity to enhance multi-agent learning by incorporating human-like *communication strategies*. The aim here is to learn what is important to communicate, not to mimic human communication. With advancements in natural language processing (NLP), multimodal AI, and object detection a wide set of tools is now available, and we propose using them for leveraging human knowledge to mitigate the challenges induced by communication. We propose a framework for a simple and efficient injection of human knowledge, which can greatly improve the performance by having a good-enough strategy to begin with, and reducing the inherent non-stationarity.

**Our contributions:**

- We propose a comprehensive framework for incorporating human-like communication strategies to enhance multi-agent learning, making it possible to solve complex tasks.
- We demonstrate the effectiveness of our framework on two *complex* multi-agent tasks and provide insights into the emergent communication behaviors observed during training.
- We publish our code and environments for further research in the field.

## 2 RELATED WORK

The non-stationary nature of cooperative MARL problems is both challenging and interesting, thus gained some attention recently. Many existing approaches embrace the centralized-learning paradigm, which enables better performance. In Foerster et al. (2016), the authors present a method for learning across agents by propagating gradients through a communication channel. The paper Sukhbaatar et al. (2016) introduce a multi-agent communication model that uses a continuous vector to transmit messages between agents. Lowe et al. (2017) utilize the policies of the other agents when choosing an action. Jaques et al. (2019) propose a reward shaping for promoting causal influence on other agents, which requires knowing all agents' policies. Decentralized learning has also been explored in recent research, here the focus is on the communication protocols. In Lin et al. (2021), the agents use autoencoders to learn an encoding for their observation, while the encoding is communicated to the other agents. Lo et al. (2023) utilize similar concepts for encoding the joint state of all agents. Eccles et al. (2019) utilize reward shaping for motivating the agents both to change their behavior upon receiving different messages, and to send more diverse messages to represent different experiences. While using centralized critic, Jiang & Lu (2018) propose an attention-based communication model that allows agents to selectively attend to incoming messages, effectively filtering out irrelevant information. Finally, Das et al. (2019) using targeted communication, to address specific messages to specific agents.

Interpretability and the emergence of natural language has also been studied in the MARL setting. Lazaridou et al. (2016) propose a method for communication using a discrete set of symbols, which can be converted to natural language by matching emergent symbols with corresponding human labels. Although this paper mainly focuses on referential games, which are rather simple in terms of control. Havrylov & Titov (2017) use a sequence of symbols for encoding complex information, such as pictures, and use grounding to make the resulted encoder more similar to a natural language,

which induce similarities but not necessarily preserve the meaning of words. Karten et al. (2023) propose a three-phase learning, where agents first learn an emergent communication protocol, then, uninformative messages are pruned, and the final phase involves teaming up with human players. On the other hand, Kottur et al. (2017) show that emergent language of artificial agents is less likely to assimilate natural language without additional constraints.

Our work is built upon these previous approaches; by learning across agents (Foerster et al., 2016), use communication as a mapping of the observation (Lin et al., 2021), filter out irrelevant information (Jiang & Lu, 2018), and shaping the reward (Eccles et al., 2019; Jaques et al., 2019). In addition, our work introduces a novel framework that combines human-knowledge with RL and could be applied jointly with (almost) any other method for our setting, to enhance its performance and increase its interpretability.

## 3  BACKGROUND AND PROBLEM SETTING

Decentralized Partially Observable Markov Decision Process (DEC-POMDP), as introduced in Bernstein et al. (2002), describes a framework in which multiple agents need to apply a decentralized policy, based on each agent's observation independently. Here, the reward function and the transition kernel operate over the joint policy of all agents, and the partial observability may extend to the centralized setting (i.e., a decentralized POMDP is a DEC-POMDP). A popular framework to deal with the challenges arising from the decentralized approach is using a communication channel, where agents share information regarding their observations and future actions to result with a better overall policy.

Formally, consider a standard DEC-POMDP: $\left(\mathcal{I}, \mathcal{S}, \{A_i\}_{i \in \mathcal{I}}, P, R, \{\Omega_i\}_{i \in \mathcal{I}}, \{O_i\}\right)$, where $\mathcal{I}$ is the set of agents, $\mathcal{S}$ is the state space, $A_i$ is the action space of agent $i \in \mathcal{I}$, $P$ is the global transition dynamics, $R$ is the global reward function, $\Omega_i, O_i$ are (respectively) the observation-space and the conditional observation probabilities of agent $i \in \mathcal{I}$. Additionally, Let $n = |\mathcal{I}|$ be the number of agent in the environment. Where a few formulations exist, adding communication to this setting can be reduced to an equivalent DEC-POMDP with increased observation-spaces (due to the communication signals) and additional action-spaces (for the communication-policy). In our setting, communication is allowed under the following condition: at time-step $t$ where an agent observes $o_t$ (the current observation) and $C_{t-1}^i, i = 1, \ldots, n$, the received communication (from all agents), it needs to choose both $a_t$, the action for the environment, and $C_t$ the communication signal that would be available for the other agents at the next time-step $t+1$. That means that $C_t$ could only depend on information the agent has at time-step $t$, hence the receiver obtains the information in delay of a single time-step.

While communication may help improve the joint policy by coordinating the agents actions and allow mitigating the partial observability that originates from the decentralized setting, it poses a major challenge – the decision-making problem of which messages should be communicated, and how to use them. Many prior works utilize a discrete communication channel, which is similar to the communication form of human beings, and can be used to decipher the message transaction. However, without additional constraints, communication may greatly differ from human-communication (Kottur et al., 2017), making it hard to interpret, even if it performs well in the given task. Similarly to Kilinc & Montana (2018), we view the communication as a mapping from one agent's observation to the transmitted message, this allows the agent to choose when to send a message, while the message itself is a continuous vector. In this case, the actual messages are expected to be relatively stationary and lossless, in terms of information contained within the original observation. In this setting, ignoring the time delay of each message, all agents would potentially have a joint observation. More formally, each agent $i$ observes its own observation at time $t$ $o_t^{(i)}$ and the broadcast communication channel $c_t$, where $c_t$ is a concatenation of $\{\phi^j(o_{t-1}^j)\}_{j=1}^K$, where $\phi^j$ is a mapping from the observation space to some vector field.

Importantly, even without explicit communication, agents could learn a well-coordinated behavior through implicit communication. Directed by this phenomenon, we formulate our testing environments (Section 5) to minimize implicit communication. We found that extreme partial observability mitigate such behavior, in particular, omitting any direct connection between the reward and the

decentralized observation. For example, switching the observations of two agents, so each one observes the other's position instead of its own (but controls its own movements).

# 4 FRAMEWORK

In this section, we present our proposed framework. Section 4.1 describes how we define and implement a simple but efficient text-to-mask model, which we use to link between humans and agents and define our communication protocol. At Section 4.2 we describe the model's dedicated architecture that allows a convenient collection and utilization of human knowledge. Then, Section 4.3 explains how we obtain a human-strategy, and how the model is trained end-to-end.

## 4.1 HUMAN-LIKE COMMUNICATION STRATEGIES

When facing a new task, humans often communicate with each other quite effectively. This happens thanks to an already existing form of communication protocol (i.e., language), an agreed terminology, and prior knowledge of the task, that allows the players to communicate well. Generally, humans have an object-oriented perception, and a task's terminology usually refer to a textual description of objects and their states. This allows to focus on a few relevant objects when communicating information, to avoid misunderstanding. While Humans determine the relevancy of an object from the task's description, prior-knowledge and previous biases, artificial agents can not directly interpret it, and it is unclear how to generally embed them. Similarly to humans, artificial agents can benefit from more focused communication (i.e., less uninformative features), but learning a 'human' communication-policy from demonstrations or including feedback in the learning process (RLHF) is likely to be unrealistic; the dimensions of MARL problems are relatively high, which would require either collecting a very large dataset of demonstrations or requesting human feedback over huge amount of simulations. We propose a hybrid approach, that combines the ability of humans to identify relevant objects with the ability of artificial agents to process the information with high dimensions.

To ground the observations, we rely on a task-dependent component, **text-to-mask**, which maps between a textual description of an object to a mask $m \in \{0, 1\}$ in the same dimensions of the (decentralized) observation-space. Potentially, the text-to-mask model may use both the textual description and the current observation to calculate the mask, for example, in case of image observations, a mask can be a segmentation of the desired object. In the general vector case, it is natural to view the observation as a collection of feature-sets, each one corresponding with a textual term that represent an object. Formally, we define a text-to-mask model $\mathcal{F} : \mathcal{T} \times O \to \{0, 1\}^{|O|}$, where $\mathcal{T}$ is the textual input space, $O$ is the observation space of each agent, and $\{0, 1\}^{|O|}$ is a binary mask space with the same dimensionality as $\mathcal{O}$. While most environments have a description of the features (e.g., angle, velocity, position, etc.) that could be used for constructing $\mathcal{T}$, complex settings may require a tailored set of terms. For image observations, it is possible to use existing pretrained object detection models to extract the mask and additional features. In our implementation, we use a set of terms, each corresponding with a fixed set of elements of the observation space, which defines a mask, (i.e., $\mathcal{F} : \mathcal{T} \to \{0, 1\}^{|O|}$). Learning from demonstrations can be fairly simple, as this model only need to construct a mask for the current observation and textual input, which does not involve control.

## 4.2 MODEL ARCHITECTURE

In this section, we elaborate our model architecture and how it allows both high expressiveness and efficient injection of human knowledge. As many other works, we used a decoupled action-space, one for the control-policy $\pi_\theta^{cont}$ and another for the communication-policy $\pi_\phi^{comm}$ (parametrized by $\theta, \phi$ respectively), where each agent implements both policies. While $\pi^{cont}$ is defined over the original action-space $A_i$, $\pi^{comm}$ in our setting correspond with choosing any subset of $\mathcal{T}_i$, which we parametrize with $|\mathcal{T}_i|$ i.i.d. Bernoulli random variables, each one indicates whether a term $\tau \in \mathcal{T}_i$ is included in the chosen subset. This subset is mapped to a mask $m$ via the text-to-mask model, and a dedicated encoder $G_\psi$ (parametrized by $\psi$) computes the transmitted communication as follows: $C^i = G_\psi(m \odot o)$, where $\odot$ stands for element-wise multiplication and $o \in \Omega_i$ is the current observation of the agent. $C^i \in \mathcal{C}$ is the transmitted communication of agent $i$, and contains informa-

tion about the agent's observation (to be received in the following time-step). Since all agents send their own communication, the policies $\pi^{cont}, \pi^{comm}$ are exposed to the current observation $o$ and an additional vector of concatenated incoming communication signals from all agents $C^{1:n} \in \mathcal{C}^n$, where $C^{1:n} = (C^1, \ldots, C^n)$. As depicted in Fig. 1a, the number of encoder instances depends on the number of policy networks, and in most cases can be reduced to two; $\pi^{cont}, \pi^{comm}$. This means that the actual communication bandwidth is doubled, although it allows the incorporation of differential communication (Foerster et al., 2016).

We incorporate a similar idea to DIAL (Foerster et al., 2016) to propagate policy gradients from the loss of the receiving agent to the encoder of the transmitting agent. It is likely to aid the policies to extract information from the other agents' observations, while we can enforce any architecture on the encoder to adapt to many types of communication protocols. $\mathcal{C}$ can be discrete, or continuous with a small dimension, the only requirement is that the encoder would be differentiable to allow gradient propagation. Propagating gradients through

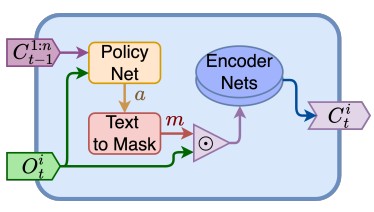 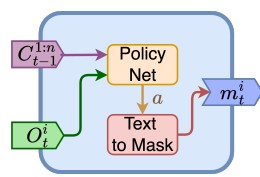

(a) Inference architecture of the communication policy. The masked observation is passed through the encoder networks of the communication-policy and control policy, then communicated.

(b) Training architecture of the communication policy. The mask corresponding with the chosen action is returned, instead of the communication signal.

Figure 1

communication requires a change of architecture during training and inference. For $\pi^{comm}$, its inference architecture is presented in Fig. 1a, but for training purposes, we use the architecture described in Fig. 1b, as we would want to train the encoder with the receiving agent. When training, $\pi^{comm}$ outputs the mask from the text-to-mask model, then it is stored along with the observation in the buffer of the receiving agent. During the training process, each policy holds an encoder (Fig. 2) which is can be trained naturally, using any policy gradient method; since we store its inputs, we can propagate gradients through the encoder. Then at inference, we switch architecture for transmitting encoded messages only. Note that the encoder operates on the masked observations of each agent separately (during training and inference), to allow decentralized execution. The high-level interconnections between the agents and policies at consecutive time-steps are presented in Fig. 3. Additionally, since we consider actor-critic architectures, we use a centralized critic as we found it to be helpful in many MARL settings (Lowe et al., 2017), which is utilized solely during the training. Note that we could use similar architecture to fine-tune the text-to-mask model as well, although in this work we consider only fixed text-to-mask models.

This architecture has a few advantages, which we shortly discuss here. Firstly, the expressiveness is unharmed, since each agent potentially obtains the information encapsulated in the other agents' observations (with a single time-step delay) and maps it to an action. Second, in terms of knowledge injection, the object-oriented view of the communication-policy is abstract enough to engineer a rule-based policy, or to realistically collect demonstrations from humans. These are what we call human-strategies, since their purpose is to determine the information to communicate rather than compute the actual communication signal, and are fairly abstract and interpretable to humans. Another desired outcome of this architecture is an enhanced interpretability: even though the communication signals can potentially be real-valued vectors with not much of a human-meaning, each one of them corresponds with a set of textual terms, which reveals the subject of the message, but not the actual content. We further discuss it in Section 7.

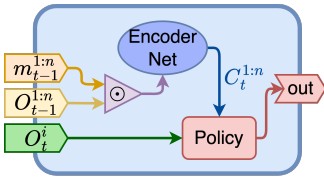

Figure 2: Training architecture of all policies. The masked observation of the transmitting agents passed independently through the encoder, which is adjusted according to the policy gradient. This applies for both the communication and the control policies.

### 4.3 TRAINING

The main difficulty that accompanies by the introduction of communication directly comes from the inconsistency of the communication-policy along the training phase; it both changes between iterations, and initially random. One approach to mitigate this inconsistency is to "cancel" the

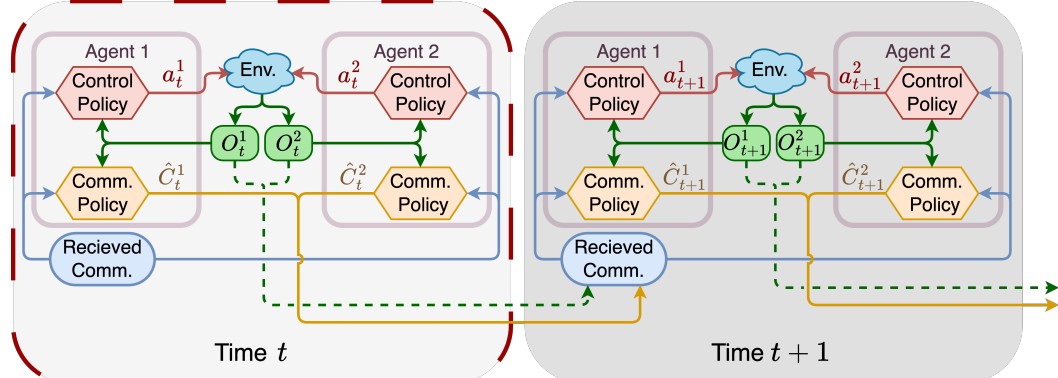

Figure 3: Interconnections between agents/policies across time-steps. The dashed lines represent the dependency on the previous observation of all agents, which holds only during the training process. Here $\hat{C}$ denotes the output of the communication-policy, whether it is a mask, or the actual signal, depends on the phase; training or inference.

communication policy, and always use a mask $m = \mathbf{1}$, although as evidenced in Section 5, in some cases this approach would fail. We propose using the human-strategies; if we have a rule-based strategy we can easily generate human demonstrations for each observation we encounter, if we only have a finite set of demonstrations available, we subsample from it to obtain a batch of human demonstrations. We utilize this batch when computing the algorithm loss by adding a BC loss constraint to the overall loss of $\pi^{comm}$:

$$\mathcal{L}_{BC}(\phi) = E\Big[-\log\big(\pi_\phi^{comm}(a^{HS}|o, C^{1:n})\big)\Big] \tag{1}$$

where $a^{HS}$ is the human demonstration. The overall loss becomes $\mathcal{L}^{comm}(\phi) = \mathcal{L}(\phi) + \beta\mathcal{L}_{BC}(\phi)$, where $\beta$ is a hyperparameter that determines proximity to the human strategy, and $\mathcal{L}$ depends on the training algorithm. For $\pi^{comm}$ we penalize the immediate reward by $\alpha$(number of objects to communicate), where $\alpha$ is a hyperparameter that prevents over-communication. Although our framework can potentially be combined with any RL algorithm, Proximal Policy Optimization (PPO) (Schulman et al., 2017) is particularly well-suited for this task as it prevents the policies from deviating too drastically during training, helping to cope with the non-stationarity introduced by the decoupled nature of the communication and control policies. Furthermore, all the agents instances share parameters, so we always have two policies to train: $\pi^{comm}, \pi^{cont}$. We train them sequentially and batch by batch; each batch of trajectories is only stored in the rollout-buffer of one of the policies, then, after the policy is updated we sample another batch for the other policy. This way, the sampled trajectories remain "on policy", which result in a better policy-gradient estimation. The sequential training comes on the expense of unused samples, although we found it crucial for convergence.

## 5 EXPERIMENTS

Our framework was tested on two simulated environments we created, that involve control and communication with extreme partial observability, in both of them, and the optimal policy relies mainly (but not only) on the communication. We compare our method against a few baselines, with varying communication. First, pure decentralized policy ('no comm') to show the added value of communication. Second, 'human-strategy', by simply applying the rule-based human-strategy. Finally, we compare against a 'dense comm' setting, in which the observations are unmasked when communicated. The dense setting can be seen as a variant of DIAL, with different underlying RL algorithm and parametrization. Our proposed method, 'hybrid', utilizes both RL and the human-strategy to learn the communication policy. In the following section, we describe each environment and present the results. We base our RL implementation on RLLib (Liang et al., 2018), our code files are available at https://github.com/commstrategies/human_like_communication_strategies, including the following simulators and the entire architecture implementation.

## 5.1 One-Dimensional Coordinate

Coordinate is a simple game. As depicted in Fig. 4 each agent has its own 1-dimensional grid world of size $k$ (in our experiments $k = 6$) that contains a single goal, which is either *shared* or *private*. At each time-step, the agents can choose to do nothing, move one step left or right (moving towards the edge results in staying put), 'claim goal', or 'mark' their location. Claiming a goal is possible when the agent and the goal are in the same location, by using the 'claim goal' action. A shared goal requires all other agents to mark the location of **this** goal in **their** grid-world. After a goal is claimed, it disappears. Each agent could only mark a single location at a given time, marking a new spot makes the previous mark disappear. The game ends when all the goals are claimed or the episode length has reached a predefined limit.

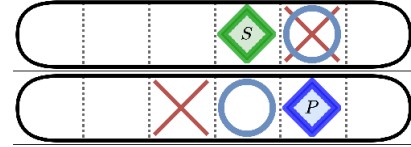

Figure 4: Snapshot from the game with two agents. The agent is the *blue* circle, shared and private goals are represented by *green* and *blue* squares respectively, and the *red* X's represent the mark of each agent.

The reward is shared across agents, although in practice we define it individually, then use its mean as the shared reward signal. The individual reward for any state and action is always $r = -2$ unless:

1. Private goal claimed: the agent who claimed the goal receives $r = 10$ (for a single turn).

2. Shared goal claimed: the agent who claimed the goal receives $r = 100$ (for a single turn).

3. Agent that already claimed a goal receives $r = 0$ when doing nothing, or $r = -5$ otherwise.

To maximize the cumulative reward, the agents should cooperate, to help each other claiming the shared-goal, while also collecting any private-goals available.

The observation-space includes the following sets of features and corresponding textual representation:

**agent**      The location of the agent on its own grid, a one-hot $k$-dimensional vector.

**goal**      One-hot $k$-dimensional array that represents the private goal location. If the goal is of shared type, it becomes **0**.

**shared goal**  One-hot $k$-dimensional array that represents the shared goal location. If the goal is of private type, it becomes **0**.

**mark**      One-hot $k$-dimensional array that represents the mark location. If the agent has not marked yet, it becomes **0**.

**achieved**  Boolean, *true* if the goal is already claimed, else *false*.

In our experiments, we use two agents. The definition of the observation-space provides a natural text-to-mask mapping. Here we use the following rule-based policy as our human-strategy: each player always communicates its own position, the position of the shared-goal if the agent's goal is of shared type, and the 'achieved' indicator if the goal (of any type) already achieved. In addition, we send the mark location is the mark is showing somewhere. We collected a set of 1000 demonstrations following this rule-based policy as our human-strategy input, and sample from it when computing the BC loss term. For this experiment, we use a control-policy with memory (incorporates recurrent neural network), and a memoryless communication policy.

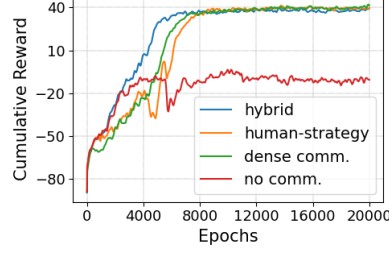

Figure 5: Results in the Coordinate domain. Our method ('hybrid') learns faster than the other baselines. With no communication ('no comm'), the agents fail to learn a meaningful policy.

We present the results in Fig. 5, we measure the mean cumulative reward over 5 random seeds. Our method along with the 'dense comm' and the 'human-strategy' eventually learn an optimal behavior, although our

proposed method seems to converge faster, supporting the hybrid architecture. Without communication, the 'no comm' method can not behave optimally due to the absence of critical information.

## 5.2 Coordinate Images

We visualize this task as a board game, each player (out of $n$ total players) has a random photo on their forehead; in it, one or more objects from a known set of objects (e.g., car, dog, etc.) is showing. While each player can not see the photo on their own forehead, they can observe the photos of the other agents. Additionally, on the board there are $k$ down-faced photos, which are enumerated, and each player has a unique coin to mark one of the $k$ photos at a time. At each turn, all players have to decide which photo they want to mark for the next turn, then mark their locations altogether and discretely observe the photo they marked. Initially, at the first turn, the coins are in the players' hands, not marking any photo. During the game, it is possible that more than one player would mark the same photo. The goal is that at the *same time*, *all players* will mark a photo that shows at least one object that appears in the photo on their forehead, where no other player is marking it. While this task appears simple for human players that communicate, it is almost unsolvable without communication.

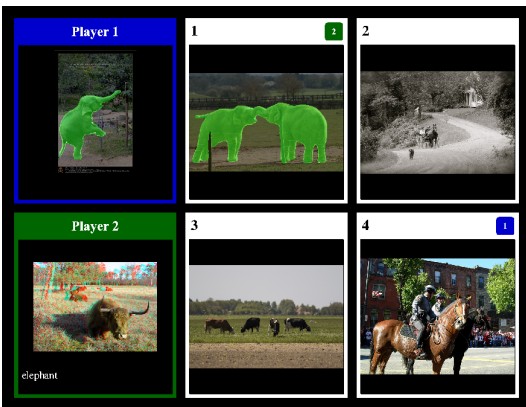

Figure 6: Snapshot from the Coordinate Images simulation where $n = 2, k = 4$. The board-photos are framed in white, and the forehead-photos are color framed. The board-photos currently chosen by a player are marked. In this example, we observe that player 2 sends the features corresponding with the word 'elephant'. For visualization purposes, we highlight the objects that are communicated. Only photos that have been viewed at times $0, \ldots, t$ are highlighted, this includes the other player's forehead photo.

Specifically, in our environment the photos and their tags (set of objects) derived from Lin et al. (2014), which is mainly used for object detection, where we chose photos that are tagged with one or more of the following 10 tags: 'car', 'airplane', 'bird', 'cat', 'dog', 'horse', 'sheep', 'cow', 'elephant', 'cake'. We process the dataset to construct a set of $9,742$ photos that attributed only with these tags.

All the photos in the game are selected randomly at the beginning of each episode, assuring that a solution exists: after the forehead-photos are chosen, $n$ photos with matching objects are randomly selected as board photos, then the remaining $k-n$ (we require $n < k$) photos are sampled from without constraints. After selected, each photo is passed through a pretrained YOLOv8 model (Jocher et al., 2023) to extract a vector $\boldsymbol{v} \in \{0, 1\}^{10}$ that indicates which objects appear in the photo. Each observation of a single agent is composed of the following:

1. One-hot vector that indicates which player is observing (e.g., for two players, $(1, 0)^T$ represents player 1, and $(0, 1)^T$ represents player 2).

2. Vectors that represent each photo, except for the forehead photo of the agent who observes: $n - 1 + k$ vectors, each $v \in \{0, 1\}^{10}$.

3. A matrix $\boldsymbol{W} \in \{0, 1\}^{n \times k}$ that indicates which photo each agent is currently marking.

4. A vector $\boldsymbol{u} \in \{0, 1\}^k$ that indicates which board-photos has been previously marked by the observing player.

These components are flattened and concatenated to construct a vectorized observation. In addition, it provides an optional centralized observation, that includes the vector representation of all the photos in the current game, along with player markings and previously viewed photos of each player. We utilize this centralized observation during the training to train the critic only.

While the agents observe the vector representations as predicted by YOLO, the environment utilizes the true tags, from the dataset, for the reward and dynamics computation. It is important since YOLO makes mistakes, which are approximately $10\%$ false positives and $8\%$ false negatives. These

mistakes amplify the partial observability, forcing the agents to compensate for YOLO. The reward is shared, and defined by the number of agents that hold the following conditions: (1) they do not co-mark a photo with another agent, and (2) they mark a photo that shares an object of their forehead-photo. Let this number be $\mu_t$, then the global reward is defined by

$$R(\mu_t) = \begin{cases} 5 & if \ \mu_t = n \\ \frac{\mu_t}{n} - 1 & otherwise \end{cases}$$

Note that $\mu_t = n$ corresponds with reaching the goal. Upon reaching the goal, the current episode ends and a new one starts. The text-to-mask model is trivial here, for a given term representing an object of the possible 10, its corresponding mask should only pass the fixed features corresponding with the specific object, which are the same row in each vectorized photo. The environment is depicted in Fig. 6. In the experiment, we use $n = 2, k = 5$ and force each episode to end after 25 time-steps.

Both policies (control, communication) in this experiment are memoryless and depend on the current observation and incoming communication only. The human-strategy we use in this experiment is to communicate all the objects that show in the other players' foreheads at each time-step. We repeat the experiment for 2 random seeds, and present the results in Fig. 7. Similarly to Fig. 5, the 'hybrid' approach outperforms the other methods, while except for the pure 'human-strategy' setting, the baselines fail to converge at all and perform similarly to a random policy. It is expected for the 'no comm' baseline to perform poorly, although the 'dense comm' method should be able to perform well, which is not the case here.

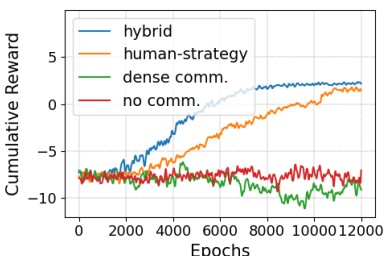

Figure 7: Results in the Coordinate Images domain. Surprisingly, the 'dense comm' setting performs poorly, similarly to the 'no comm' setting. The methods that mask the communication ('hybrid', 'human-strategy') manage to learn a successful behavior. As expected, 'no comm' shows that without communication, the task become unfeasible.

## 6 INTERPRETABILITY AND EMERGENT BEHAVIOR

Our framework and model are designed for convenient transfer of human-knowledge to artificial agents (Section 4.1). However, for interpretability, we need to transfer knowledge in the other direction – from artificial agent to humans. As discussed earlier, while the resulted communication is not directly grounded to natural language, the context is – *what the agents are talking about*, instead of *what are they saying*. The context along with the agents' behavior can provide a wider understanding on their policy and what are they actually communicating.

In Fig. 8, we present a sample trajectory from the resulted 'hybrid' method, in the Coordinate simulation (Section 5.1). For every time-step, we show the state and the objects each agent chose to transmit. Here, One agent has a private-goal, and the other have a shared-goal. Initially, the agent at the top communicates its own position and the position of the shared-goal, while the other (bottom) agent only communicates its own position. Over the next time-steps, the top agent moves towards the shared-goal, then waits for the other agent. The bottom agent marks and claims its private-goal, then moves on to mark the position of the top agent's shared-goal. During this period, agents only communicate their position. Once marked, the bottom agent communicates the mark's position, and the 'achieved' status, then, the shared-goal is claimed and the simulation ends.

Analyzing the agents' behavior, we can learn a few things: (1) Since the shared-goal position is communicated once at the beginning, while the bottom agent knew exactly where to mark, we can deduce that the policy indeed utilize its memory. (2) After the bottom agent marks the shared-goal position, it immediately communicates both the mark's location and his 'achieved' status. This can be attributed to the bottom agent rushing the top agent to claim his goal – "I already claimed mine, go ahead and claim it". (3) The agents communicate their location at all times, except the bottom agent, once marked the shared-goal. This information can be used to determine which goal to claim first, and once an agent finished their task, their location is irrelevant anymore.

Although it is a rather simple environment, this form of communication provides a specific context for a human "side-listener" that observes the agents' behavior, and could be used in environments

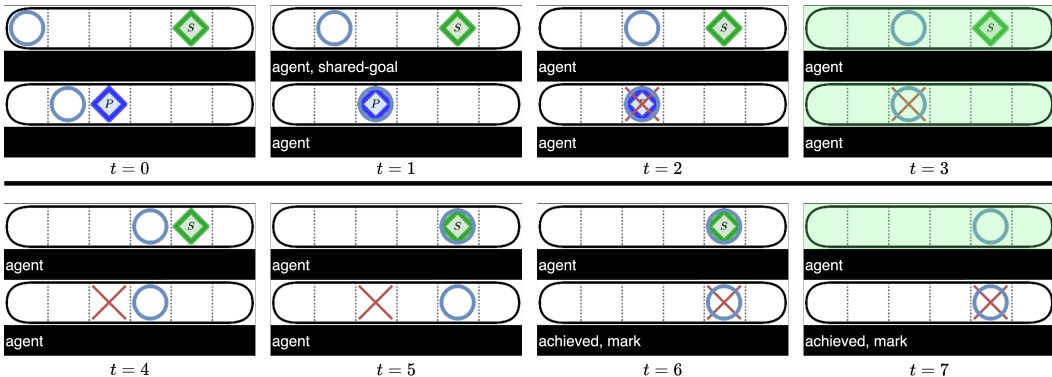

Figure 8: Sample trajectory from Coordinate (Section 5.1), it shows the transmitted communication at each time-step. The communication provides a glimpse to the agents' decision-making mechanism, by analyzing the information exchange, and the following decisions of each agent.

that are much more complex. Moreover, the main component that enables this interpretability is the text-to-mask model, which is easier to obtain for human inputs, such as images, due to the wide availability of pretrained models.

## 7 DISCUSSION AND FUTURE WORK

Communication can potentially improve the performance in any MARL task, but in practice it is very hard to learn effective communication protocols, especially in dynamic settings that involve partial observability, where efficient communication is critical. However, there is no free lunch, and to learn efficient communication we turn to the masters – humans. Our abstraction of the object-oriented perception derived an architecture that with a minimal engineering effort could be adapted to many domains, extracting and utilizing human knowledge to enable effective communication. Moreover, as we demonstrate in Section 6, we are able to observe the context of a conversation, which helps to interpret and even communicate with the agents.

We believe that hybrid approaches, such as ours, that incorporate human-knowledge in RL pave the path for practical applications in many domains that are perceived to be hard for artificial agents. One major issue that arises when involving humans in the training is the data-collection process. Especially in RL, collecting human demonstrations is not trivial and in many cases impractical. On the other hand, smart architectures and framework that consider this issue may be utilized, such as in this work.

Further research could be done in several directions. First, we use a simple text-to-mask model, which could be replaced with a more sophisticated LLM-based model, in light of the recent developments in this field. Second, more complex textual descriptions may require an impractical action-space for the communication policy, this could again be addressed with an LLM. Third, the concept of policies that mask other policies observations could be extended to the communication times (i.e., policies that determine at what times to communicate if at all), or applied in a single-agent setting, where the observation-space is large or contains many distractions. Finally, our framework allows humans to intervene during the training or even through the inference by communicating various observations over the communication channel, although it is possible, it is outside our scope.

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
