# OpenReview forum: "Human-like Communication Strategies for Improved Multi-Agent Reinforcement Learning"
_ICLR.cc/2025/Conference — ICLR 2025 Conference Withdrawn Submission_

### Official Review · Reviewer_Ebrf · 2024-10-29

**Soundness:** 1
**Presentation:** 1
**Contribution:** 1
**Rating:** 1
**Confidence:** 5

**Summary:**

This paper attempted to improve the communication protocol in MARL. To this end, the authors proposed a text-to-mask strategy to encode observations for communications among agents.

**Strengths:**

Except that the research question at the beginning looks interesting, I cannot find any strengths in this paper.

**Weaknesses:**

1. This paper is not well-written, there are lots of typos. For example, centralized -> decentralized in line 128, need -> needs on line 202, is can be -> can be on line 242, would want to -> would like to on line 239, etc.
2. As for line 162 that "For example, switching the observations of two agents, so each one observes the other’s position instead of its own (but controls its own movements)," I have some different viewpoints. Is it still valid when the reward function is a invariant to observations or agents' observations are symmetric? In either case, we can still always find a relation between reward function and observations.
3. About line 193-195 that "In the general vector case, it is natural to view the observation as a collection of feature-sets, each one corresponding with a textual term that represent an object," why can this form of observation representation can enhance identification of the relevancy of an object?
4. As for line 199-200 that "For image observations, it is possible to use existing pretrained object detection models to extract the mask and additional features," even if the proposed text-to-mask is good representation, the inaccuracy brought up by the pretrained feature extractor can still lead to some unexpected errors. In turn, the embedding is still uninterpreted.
5. As for line 208-210 that "As many other works ... implements both agents," why do you choose the decoupled policies for selecting actions and communications, respectively? If there is no justifiable reasons, this setup would only result in more parameters.
6. As for line 237-239 that "For $\pi^{comm}$ ... with the receiving agent," why is this reasonable? Is there any special considerations behind this design?
7. As for line 243-244 that "Then at inference ... messages only," is this a logical reason to explain why encoder network is disregarded? Could you give more clarification about this?
8. As for line 289-292 that "We propose ... demonstrations," how is the rule-based policy or human demonstrations be guaranteed to find for every task? Also, the use of BC contradicts the claim in line 65-67 that "Another appraoch would be ... which is hard to collect."

**Questions:**

Please clarify the concerns in weaknesses.

---

### Official Review · Reviewer_vwzZ · 2024-10-31

**Soundness:** 2
**Presentation:** 2
**Contribution:** 1
**Rating:** 3
**Confidence:** 4

**Summary:**

The paper proposes a framework that incorporates human-inspired communication strategies to enhance coordination and learning in multi-agent reinforcement learning (MARL). The key motivation lies in addressing the challenges posed by communication in MARL, such as increased non-stationarity and large state space dimensionality, by drawing parallels to how humans communicate efficiently using contextual awareness and prior knowledge. The authors introduce a hybrid communication model that leverages text-to-mask mapping and human strategies, combining rule-based human demonstrations with reinforcement learning techniques. The framework is evaluated on two distinct multi-agent environments, demonstrating its effectiveness in improving both communication protocols and overall task performance. Additionally, the paper provides insights into emergent communication behaviors observed during training and emphasizes the potential for improved interpretability through human-like communication protocols.

**Strengths:**

The paper combines human-inspired strategies with reinforcement learning, addressing a key challenge in multi-agent reinforcement learning (MARL): developing effective communication protocols.

**Weaknesses:**

The proposed framework mainly integrates existing methods such as text-to-mask models and PPO, offering limited original theoretical contributions.

The experiments are conducted in relatively simple environments, limiting the framework’s demonstrated applicability to more complex real-world MARL scenarios.

The analogy to human communication is weak, as the proposed strategies focus on rule-based data exchange rather than capturing the depth of real human interactions.

The paper overlooks a deeper exploration of communication policy instability during training, which could pose challenges in dynamic environments.

While the related work is discussed, the experimental section lacks comprehensive benchmarks against state-of-the-art communication strategies, making it difficult to assess the framework’s comparative advantage.

**Questions:**

Could the authors provide more direct comparisons between their hybrid framework and other recent methods?

---

### Official Review · Reviewer_Q4s7 · 2024-11-03

**Soundness:** 2
**Presentation:** 2
**Contribution:** 2
**Rating:** 5
**Confidence:** 4

**Summary:**

Communication in Multi-Agent Reinforcement Learning has shown promise in enabling better coordination in complex environments. Yet, it faces challenges like non-stationarity and interpretability. This paper proposes a more human-like communication approach. More specifically, it uses text-to-mask models which map descriptions of an object to a mask. The communication protocol is learned via DIAL with human demonstrations. Such object-oriented and mask-based messaging makes it relatively straightforward to leverage human demonstrations, with higher interpretability. The method’s effectiveness is demonstrated on two newly proposed multi-agent tasks.

**Strengths:**

- To the best of my knowledge, such an object-oriented based messaging approach is relatively novel to a small extent
- The paper in general is comprehensively written

**Weaknesses:**

- The consideration of ‘complex’ problems in the introduction is the same as the cheap talk discovery problem (e.g., the phone booth maze) described in [1]. Please cite it
- The approach could be better motivated in the introduction by being more specific about which aspects of human-like communication are being considered here
- With only 2 agents with 2 relatively simple environments, the method's effectiveness could be better supported. It should also be compared against other MARL communication baselines like AEComm (Lin et al., 2021).

[1] Lo, Y. L., de Witt, C. S., Sokota, S., Foerster, J. N., & Whiteson, S. Cheap Talk Discovery and Utilization in Multi-Agent Reinforcement Learning. In The Eleventh International Conference on Learning Representations.

**Questions:**

- The method seems a bit limited and hard to generalize given the need to hand-design the message space. Can you provide some elaboration on this?

---

### Official Review · Reviewer_yEKm · 2024-11-09

**Soundness:** 2
**Presentation:** 1
**Contribution:** 2
**Rating:** 3
**Confidence:** 3

**Summary:**

The paper presents a framework to integrate human-like communication strategies into multi-agent reinforcement learning (MARL) to enhance performance in complex tasks. Inspired by human communication, the authors incorporate a "text-to-mask" model to focus on relevant features in observations and use imitation to human demonstrations as an auxiliary objective. The proposed approach, which combines reinforcement learning (RL) and rule-based strategies, is tested in environments with high partial observability. Results suggest that the hybrid approach achieves faster learning and improved interpretability in communication behaviors.

**Strengths:**

* The paper introduces a unique combination of human-inspired communication and RL, tackling the challenges of MARL in complex environments.
* The "text-to-mask" model facilitates interpretability, allowing communication to be contextually grounded, making agent interactions more understandable.

**Weaknesses:**

* The paper is not very well written. There are several instances of grammatical mistakes, informal wording and poor sentence formation. Some examples include:

    - Line 58: "it could be beneficial assimilating to human beings,..." (assimilating could be replaced by a more appropriate word)
    - Line 87: "problems is both interesting and challenging, thus gained some attentions..." (informal language)

    Fixing these errors and sentence formation issues would improve the paper significantly

* The introduction of the paper does not do a very good job at motivating the approach. Elaborating upon human communication strategies such as prior knowledge, contextual understanding and efficient information exchange with specific examples of how they are useful in human context and how they could be useful in the context of multi-agent RL would help motivate the approach better. Section 4.1 does attempt to do the above, partially, however, improving its sentence formation and incorporating it in the Introduction would strengthen the introduction.

* The approach is not compared against standard MARL baselines such as IPPO [1], MAPPO [2], QMix [3],

* The experiments are performed on rather toyish and non-standard environments. Validating the results on standard environments (for eg. environment from OpenAI's Multi Particle Environment suite) and environments with higher dimensional observation spaces (such as Starcraft) would strengthen the paper significantly.

* The authors do not provide any hyperparameter details.

References:

[1] Chao Yu, Akash Velu, Eugene Vinitsky, Yu Wang, Alexandre Bayen, and Yi Wu. The surprising effectiveness of ppo in cooperative, multi-agent games. arXiv preprint arXiv:2103.01955, 2021.
[2] Christian Schroeder de Witt, Tarun Gupta, Denys Makoviichuk, Viktor Makoviychuk, Philip H. S. Torr, Mingfei Sun, and Shimon Whiteson. Is independent learning all you need in the starcraft multi-agent challenge?, 2020.
[3] Tabish Rashid, Mikayel Samvelyan, C. S. D. Witt, Gregory Farquhar, Jakob N. Foerster, and Shimon Whiteson. Qmix: Monotonic value function factorisation for deep multi-agent reinforcement learning. ArXiv, abs/1803.11485, 2018

**Questions:**

* Can the authors provide experimental results on some standard environment such as MPE and Starcraft?
* How does the notion of "simple" and "complex" environments tie into the tasks used in the experiments section. Would the tasks used be considered simple or complex? And why?

---

### Note · Authors · 2024-11-17

**Comment:**

We thank the reviewers for their time and effort, as well for the helpful comments. We have decided to withdraw our paper at this time.

**Withdrawal Confirmation:**

I have read and agree with the venue's withdrawal policy on behalf of myself and my co-authors.